# A Novel Workflow with a Customizable 3D Printed Vaginal Template and a Direction Modulated Brachytherapy (DMBT) Tandem Applicator for Adaptive Interstitial Brachytherapy of the Cervix

**DOI:** 10.3390/jcm11236989

**Published:** 2022-11-26

**Authors:** James J. Sohn, Mitchell Polizzi, Dylan Richeson, Somayeh Gholami, Indra J. Das, William Y. Song

**Affiliations:** 1Department of Radiation Oncology, Northwestern Memorial Hospital, Northwestern University Feinberg School of Medicine, Chicago, IL 60611, USA; 2Department of Radiation Oncology, Virginia Commonwealth University, Richmond, VA 23219, USA

**Keywords:** direction modulated brachytherapy (DMBT), 3D printing, image-guided adaptive brachytherapy

## Abstract

A novel clinical workflow utilizing a direction modulated brachytherapy (DMBT) tandem applicator in combination with a patient-specific, 3D printed vaginal needle-track template for an advanced image-guided adaptive interstitial brachytherapy of the cervix. The proposed workflow has three main steps: (1) pre-treatment MRI, (2) an initial optimization of the needle positions based on the DMBT tandem positioning and patient anatomy, and a subsequent inverse optimization using the combined DMBT tandem and needles, and (3) rapid 3D printing. We retrospectively re-planned five patient cases for two scenarios; one plan with the DMBT tandem (T) and ovoids (O) with the original needle (ND) positions (DMBT + O + ND) and another with the DMBT T&O and spatially reoptimized needles (OptN) positions (DMBT + O + OptN). All retrospectively reoptimized plans have been compared to the original plan (OP) as well. The accuracy of 3D printing was verified through the image registration between the planning CT and the CT of the 3D-printed template. The average difference in D_2cc_ for the bladder, rectum, and sigmoid between the OPs and DMBT + O + OptNs were −8.03 ± 4.04%, −18.67 ± 5.07%, and −26.53 ± 4.85%, respectively. In addition, these average differences between the DMBT + O + ND and DMBT + O + OptNs were −2.55 ± 1.87%, −10.70 ± 3.45%, and −22.03 ± 6.01%, respectively. The benefits could be significant for the patients in terms of target coverage and normal tissue sparing and increase the optimality over free-hand needle positioning.

## 1. Introduction

Plan quality for cervical cancer brachytherapy has improved steadily since the introduction of image-guided adaptive brachytherapy (IGABT) workflow where patient-specific target volume is defined and its dosimetric conformality evaluated on a per-fraction basis [1,2]. The recommendation by Groupe Européen de Curiethérapie–European Society for Therapeutic Radiology and Oncology (GEC-ESTRO) on the use of volumetric imaging (particularly magnetic resonance imaging (MRI)) [3] has led to rapid developments and updates to the common gynecological (GYN) applicators such as tandem, ovoids, ring, interstitial templates, etc. To date, the changes are largely to accommodate MR compatibility and better needle positioning through standard tracks made within the intracavitary applicators (i.e., intracavitary-interstitial (IC-IS) hybrids) and/or standard needle templates [4,5].

However, there is still room to improve. For example, there has been a steady increase in activity on the development of brachytherapy applicators that conforms to the intensity modulated brachytherapy (IMBT) paradigm [6,7]. The IMBT provides at least one additional degree of freedom in the dose delivery process in order to achieve a higher degree of dose conformality, i.e., the directionality of the dose profile as opposed to the standard isotropic profile, achieved through incorporation of high-density shielding materials. One such innovation is termed direction modulated brachytherapy (DMBT) [8,9,10,11,12,13,14,15,16,17,18,19]. Of which, the DMBT tandem applicator, first proposed in 2014 [10], has six symmetric peripheral grooves along a paramagnetic tungsten alloy rod (95% W, 3.5% Ni, and 1.5% Cu) with a density of 18.0 g/cm^3^ and a thickness of 5.4 mm. Due to the high density of the tungsten alloy, a highly directional radiation dose profile can be generated, as shown in Figure 1. It has also been proven to be MR compatible with negligible artifacts on 1.5 T and 3 T MRI systems [12,13]. On 75 intracavitary brachytherapy plans treated with a standard tandem-and-ring applicator, which were re-planned with the DMBT tandem-and-ring, it was shown to reduce the D_2cc_ for the bladder, rectum, and sigmoid by on average (±SD) 8.5% ± 28.7%, 21.1% ± 27.2%, and 40.6% ± 214.9%, respectively, while maintaining identical D_90_ coverage to the high-risk clinical target volume (CTV_HR_) [10]. While the achievements are noteworthy, the cases were intracavitary with a limited CTV_HR_ volume range of average (±SD) 24.3 ± 10.4 cm^3^. It is well known that larger volumes (e.g., ≥30 cm^3^) tend to require interstitial needles to reach D_90_ ≥ 85 Gy-EQD_2_, either as free-hand or as part of IC-IS hybrid applicators, as the dosimetric reach by tandem is regional (e.g., within ~3–4 cm) by the nature of the energy spectrum generated by the Ir-192 brachytherapy sources [20,21]. So, for interstitial brachytherapy cases, it becomes not only necessary to determine the number of needles to use but to place them in optimal geometric locations for best target coverage, even when the DMBT tandem is used, as the effect of intensity modulation enabled from this tandem is also regional [11,14].

Newer commercial applicators (e.g., IC-IS hybrids and interstitial templates) have improved for better needle guidance but largely have standard needle tracks, angulations, and/or depth of reach. Such may be the reasons some practitioners opt to insert needles manually due to the freedom it allows of arbitrary angulations, etc., which may best fit the case. A better solution to this, in our opinion, is the 3D printed needle-track templates that fit and conform to the patient-specific vaginal space. Studies have shown their dosimetric benefit through flexibility in needle angulations, but at the same time, allowing for the reduction in the variability surrounding the manual needle insertion process by practitioners as the needle tracks are built-in into the template itself [22].

We propose a novel interstitial brachytherapy workflow that utilizes the advantages of both IMBT and 3D printed needle-track templates, all within the framework of IGABT. We demonstrate the feasibility of producing an accurate patient-specific 3D printed vaginal template, with built-in needle tracks, for a clinical patient case. Our hypothesis was that there is potential dosimetric improvements of the DMBT tandem with spatially optimized needles that would be achievable with the 3D-printed template, on five patients treated with interstitial brachytherapy. The goal of this work is to propose an innovative IGABT workflow that pushes the boundaries of current physical limits in order to achieve unprecedented dosimetric conformality while simultaneously achieving a high degree of reproducibility in needle positioning.

## 2. Materials and Methods

### 2.1. The Proposed Workflow

The overall steps to our proposed two-step optimization, 3D modeling, and rapid 3D printing workflow are presented in Figure 2. This process can be adapted in various ways and to the preference and needs of each clinic. The proposed workflow must start with imaging acquired either pre-treatment or the day of the first fraction. The pre-treatment imaging modality, if implemented, should ideally be MRI in order to best identify the CTV_HR_ for planning [3]. This workflow would be best served with a dedicated MRI simulator integrated with a brachytherapy suite to allow for the rapid acquisition required for planning and treatment imaging. Our institution recently acquired and commissioned a new 3 T MRI simulator (Vantage Galan, Canon Medical Systems USA, Tustin, CA, USA) dedicated to our radiation oncology department and the basis of this workflow proposal is grounded in the expected growth of MRI use for cervical cancer brachytherapy treatments. However, this workflow can include CT imaging as the imaging modality of choice but would ideally transition towards MRI guidance eventually. The pre-treatment (or pre-implant) imaging requires either polymer gel or contrast-soaked vaginal packing to delineate the vaginal vault and represent the surface of the patient-specific 3D vaginal template. The number of needles and their geometric positions are manually optimized by a planning physicist based on the pre-treatment imaging, and the second inverse optimization step is to then determine the dwell times within all of the dwell positions inside the needles and the DMBT tandem applicator. The initial manually optimized planning may include inverse optimization to ensure that adequate coverage is achieved with the selected number of needles and placement, however the positioning of the needles will be placed based on patient anatomy and clinical judgement. One of the model-based dose calculation algorithms (MBDCA), described in the AAPM TG-186 report [23], should be used during the optimizations and the final dose calculations in order to account for the high-density heterogeneities (including the tungsten alloy in the DMBT tandem). Additional needle tracks may be supplemented at this point at the discretion of the physician and/or physicist to provide more robustness against geometric uncertainties such as intrafractional organ and applicator motions [24].

Once a satisfactory plan is obtained, the contour set of the vaginal cavity including the tracks of the needles, the DMBT tandem, and ovoids or ring (if included), are separately exported for the 3D modeling process. In the case that a pre-treatment imaging is not available, the first fraction’s simulation imaging can be used to perform this step with a caveat that the 3D printed template will not be available on the first fraction (same-day) treatment, so a conventional plan should be delivered on the first fraction then. The next step is to export the contour set as a stereolithographic (STL) file from the treatment planning system (TPS). The STL model post-processing and 3D printing are completed within 24 h and require minimal manual labor including sterilization process. The next treatment fraction(s) would follow the same process but includes an iterative decision loop to determine whether the 3D printed template and/or needle positions need to be further updated/adapted based on the (regressing) target and the surrounding anatomy. An example of the proposed workflow is illustrated in Figure 2.

### 2.2. A Patient Example

A previous locally advanced cervical cancer (LACC) patient treated at our institution with a tandem-and-ovoids (T&O) applicator set with two freehand-inserted needles, which would potentially benefit from the new workflow, was identified. Our institution’s general process of implant, imaging, and volume optimized planning has been reported previously [25]. Following the review of the treated plan, a separate DMBT-tandem-based plan was created using our research TPS (BrachyVision^®^ v.16.1, Varian, Palo Alto, CA, USA) and calculated by Acuros BV dose calculation algorithm. First, a 3D solid applicator model of the prototype DMBT tandem (Figure 1C—in development, under a research agreement with Varian) was inserted virtually in the same position as that of the conventional tandem used clinically. Then, the two needle positions were individually translated and rotated manually for optimal geometric coverage of the CTV_HR_ while not intersecting any organs at risk (OARs) as well as the DMBT tandem-and-ovoids (DMBT + O). Finally, an inverse optimization was initiated using the same dose-volume constraints used for the treated plan to arrive at a satisfactory final plan. Figure 3 demonstrates a side-by-side comparison of the original and the new plan. In this work, a lack of commercial needle-position-optimization algorithm in our research TPS forced us to manually optimize the positions of the two needles. However, it is our intent to develop this algorithm in the future for a more seamless implementation of the overall proposed workflow.

### 2.3. 3D Printed Vaginal Template

A 3D printed vaginal template was created based on the patient-specific vaginal space outlined from the vaginal packing, the positions of the T&O and needles, and with minor adjustments to account for replacing the conventional tandem with that of the DMBT tandem. The entire length of the vaginal canal was delineated and extended outside the opening of the vagina to allow easy handling of the template. In order to create a 3D printable file, the final contour set was exported as a stereolithography (STL) format directly from the TPS. The 3D tracks of the applicators and the needles were also exported as a comma-separated values (.csv) file and processed using an in-house developed python script to be 3D printable. The material used in the study was a United States Pharmacopoeia (USP) Class VI medical-grade photopolymer Accura Clearvue with Clear Coat (3D Systems, Inc., Valencia, CA, USA), which can be sterilized for recurrent clinical use, as necessary [26]. The 3D printed vaginal template was then scanned with our CT simulator and registered with the planning CT in order to assess volume and shape agreement, as part of a proposed quality assurance (QA) workflow. Figure 4 illustrates each step of the process. Note that in Figure 4c, the DMBT tandem prototype appears straight (i.e., not curved like a hockey stick). This was done for illustration purposes only and will be bent with standard multiple angular options (e.g., 15–60°) when implemented clinically, as shown in Figure 1D,E.

### 2.4. Retrospective Plan Comparison

Five previously treated LACC patients were retrospectively re-planned to assess the dosimetric benefit of needle spatial-position optimization. Two additional plans were created for each patient. One plan was the original ovoids (O) and needles (ND) with the conventional tandem replaced by the DMBT tandem (DMBT + O + ND). The other plan (DMBT + O + OptN) was the original ovoids with the DMBT tandem and the needles spatially optimized. The spatially optimized needles were manually rotated and translated based on the patient anatomy and were adjusted from the original positioning to ensure that their placement was clinically feasible. The number of needles was held constant between the plans. The dwell times were each inversely optimized using the research version of BrachyVision^®^ that models the DMBT tandem. The D_90_ for each plan was held closely constant (<1% difference), as listed in Table 1, and the D_2cc_ of the rectum, bladder, and sigmoid were recorded for analysis.

## 3. Results

A 3D printed vaginal template was successfully created. Figure 4A,B shows the STL model and Figure 4C shows the printed version with the needles and the prototype (straight) DMBT tandem inserted (for illustration only; not-for-clinic prototype shown). The registered template images showed a good agreement with a volume difference of less than 5% (Figure 4D). A comparison of the original and the DMBT + O + OptN plans is shown in Figure 3. As can be seen with the 100% isodose lines, both the DMBT tandem and the manually reoptimized positioning of the two needles produces marked overall improvement in dose conformality over the original clinical plan. The creation of the template and the subsequent performance of QA shown in Figure 4 can be achieved within a single day (≤24 h), which would be sufficiently rapid for its process to be completed and ready for the next treatment fraction(s) and to adapt to the changing anatomy, if so needed. The average differences in D_2cc_ for the bladder, rectum, and sigmoid between the original and DMBT + O + OptN plans were −8.03 ± 4.04%, −18.67 ± 5.07%, and −26.53 ± 4.85%, respectively, and the average differences between the DMBT + O + ND and DMBT + O + OptN plans were −2.55 ± 1.87%, −10.70 ± 3.45%, and −22.03 ± 6.01%, respectively (Figure 5). This demonstrates the benefits of further optimizing the needle positions when used with the DMBT tandem applicator, on top of the benefits already achieved by simply using the DMBT tandem over a conventional single-channel tandem. The image guided 3D printing workflow proposed in this work enables clinical implementation of the DMBT + O + OptN plan generation and delivery.

## 4. Discussion

The idea of utilizing the patient-specific vaginal space for generating personalized 3D templates or molds for interstitial gynecologic brachytherapy is powerful, but not new [22,27,28]. Our proposal, however, improves upon this idea by utilizing the DMBT tandem to the mix by intensity modulating the vicinity of the cervical canal that is within the dosimetric reach from the tandem and uses optimally positioned needles to complement the technology by covering regions of the target unreachable by the tandem. These vaginal templates are inherently patient-specific and allow for ease of implementation while providing a superior starting point for dose-volume optimization. It facilitates an unconstrained number of and positioning of the needles (i.e., allowing arbitrary angulations). In our example patient case for 3D printing and the five cases for retrospective planning analysis, we manually reoptimized the needle positions based on the geometric locations of the anatomy of the target and the OARs, and our understanding of the dosimetric capabilities and limits of the DMBT tandem, but of course, it is in our long-term interest to develop an optimization algorithm suitable for this task for a more seamless implementation of the overall proposed workflow. In fact, this is an area of active research in general [28,29,30,31,32].

Geometric needle positioning optimization would be beneficial in future iterations of the DMBT workflow, however, the current optimization was done manually. Angulation and position of needles were adjusted to account for CTV_HR_s and OARs, while limiting positions to clinically feasible placements. While there is no guarantee that the re-optimized positioning is the most optimal, the re-optimized positioning takes into account the location of the OARs and CTV_HR_, which does not occur in the current workflow that only uses ultrasound guidance or pre-procedure imaging. Re-optimization occurs manually according to the planner’s experience, but may be an iterative process in order to determine whether a different arrangement of needles can improve coverage and reduce OAR dose. For comparison with the clinical plan, the plan with the overall greatest reduction in D_2cc_ for each OAR was selected where several manually optimized needle plans were generated.

A critical distinction for 3D printing for interstitial brachytherapy is that the resin used for printing must be biocompatible, nontoxic and able to withstand repeat-sterilization. In addition, the 3D printing materials used should also not significantly affect the dose distributions. Most resins have densities ranging between 1.0 and 1.3 g/cm^3^, and for HDR brachytherapy applications should cause only minor dosimetric perturbations compared to a water-equivalent applicator [27]. For further review, see literature that has covered the effects of dosimetric perturbations caused by 3D printing materials [33,34,35,36].

Cervical cancer brachytherapy is declining in the United States [37], and some of the barriers to its adoption include lengthy procedural times and inadequate maintenance of the applicator and needle insertion skills. However, with the anticipated high-reproducibility of implant quality with accurate needle placement due to the built-in needle tracks in the vaginal template, the burden of practitioners’ experience may be significantly reduced or eased. This workflow, while straightforward, may require some work and additional cost to start implementing. Cervical cancer brachytherapy outcomes have been reported to be linked to the implant quality [38] and having a template to help guide accurate insertion of needles (to ensure an optimal dosimetric plan) is expected to provide better outcome.

## 5. Conclusions

A novel workflow presented here is one of the more adaptive approaches proposed to date. The rewards can be extensive for the patients in terms of target coverage and normal tissue sparing, but not only so, it would also reduce the variability surrounding the manual needle insertion process by practitioners since the needle tracks are built-in into the template. We demonstrated that there are significant theoretical improvements in OAR sparing with the DMBT technology compared to the conventional plans and that additional sparing is achievable through optimization of needle positions. As the field continues to trend towards MRI-guided brachytherapy with volume-based (inverse-) optimizations, it seems quite timely to begin investigating the benefits of what inverse planning with the DMBT tandem and 3D printing would potentially offer, especially, for challenging interstitial cases with extensive target volumes.

## Figures and Tables

**Figure 1 jcm-11-06989-f001:**
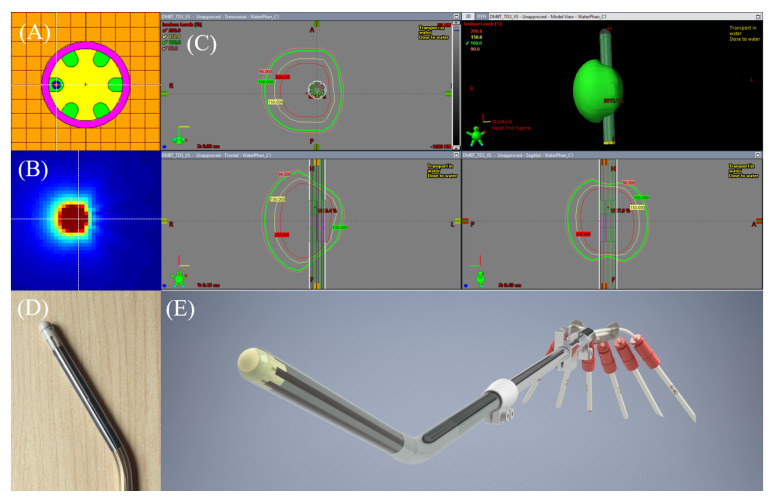
(**A**) A cross-section of the prototype direction modulated brachytherapy (DMBT) tandem applicator. (**B**) A Monte Carlo simulated directional dose profile of a clinical ^192^Ir source generated by the DMBT tandem. (**C**) The DMBT tandem 3D solid applicator model created in a research-version of BrachyVision^®^ (Version 16.1) and, as can be seen, a directional dose profile generated by the Acuros BV^®^ dose calculation algorithm in all three standard angles is illustrated (Varian, A Siemens Healthineers Company, Palo Alto, CA, USA). This research version of BrachyVision^®^ allows inverse optimization with the DMBT tandem, taking into account full heterogeneities of the applicator, as well as the combined dwell positions distributed within other applicators and needles. (**D**) A prototype DMBT tandem shown with a dummy ^192^Ir source travelling inside one of the grooves. (**E**) A 3D computer-aided design (CAD) drawing of the prototype DMBT tandem shown in (**D**).

**Figure 2 jcm-11-06989-f002:**
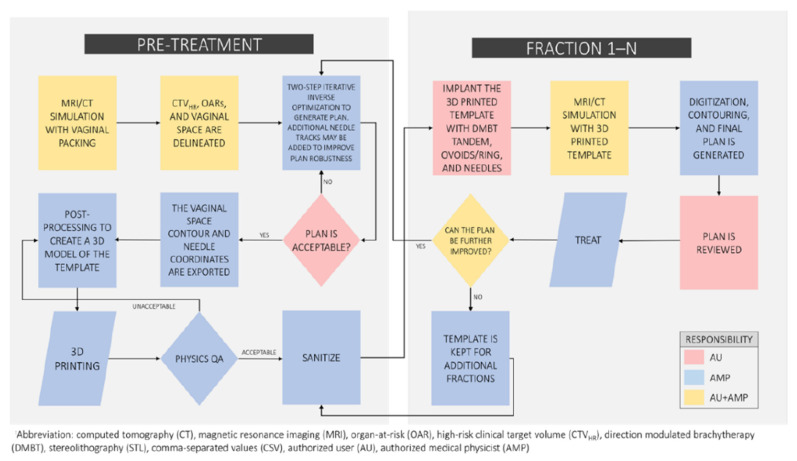
The proposed workflow with identified personnel to perform each step.

**Figure 3 jcm-11-06989-f003:**
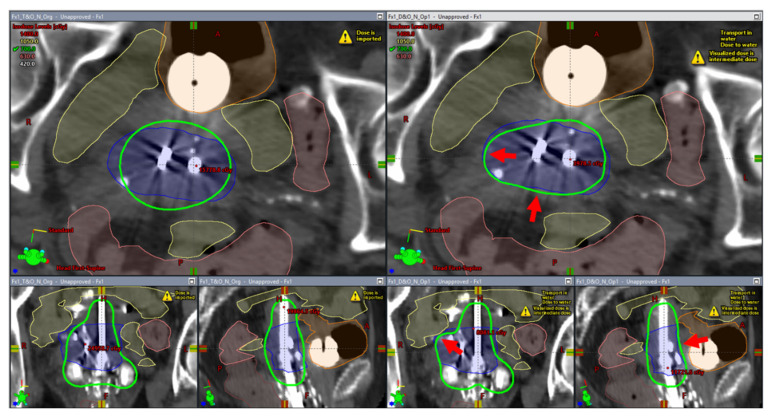
The difference in dosimetric conformality achieved by the original clinical plan (**left**) and the plan optimized with the direction modulated brachytherapy (DMBT) tandem, ovoids, and the two needle tracks (**right**). The blue contour is the CTV_HR_ while the green is the 100% isodose line. The four red arrows on the right image set indicates notable improvements in dose conformality enabled by the proposed workflow where optimal positioning of the needles (**top and bottom left**) and the intensity modulation enabled by the DMBT tandem around its vicinity (**bottom right**) are well observed.

**Figure 4 jcm-11-06989-f004:**
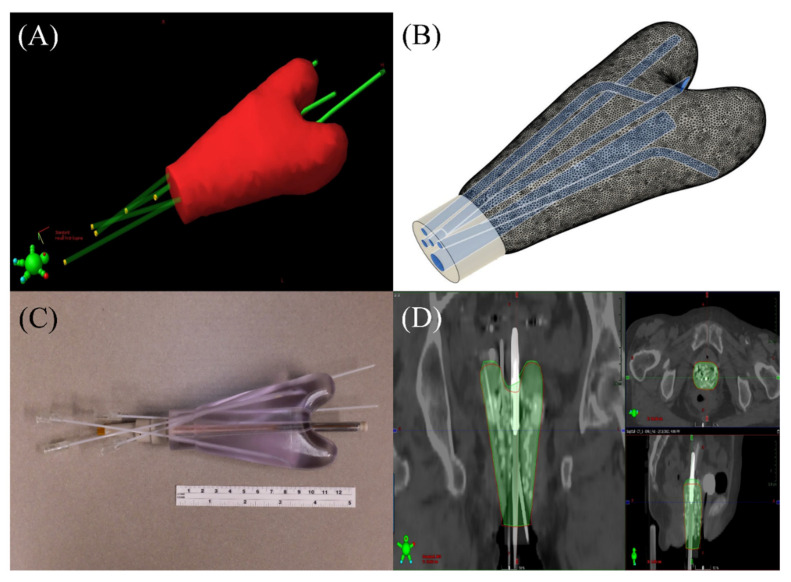
The 3D model and the printed version of the vaginal template with the two needle tracks, the two ovoid tracks, and the DMBT tandem are illustrated. (**A**) The vaginal space contour with the needles and applicator tracks are defined. (**B**) The subsequent 3D model of the template with the tracks are generated for 3D printing. (**C**) The printed 3D vaginal template with the needles and a prototype direction modulated brachytherapy (DMBT) tandem inserted for imaging and QA. (**D**) Image registration between the planning CT and the CT of the vaginal template are performed to evaluate and confirm the accurate printing of the 3D template.

**Figure 5 jcm-11-06989-f005:**
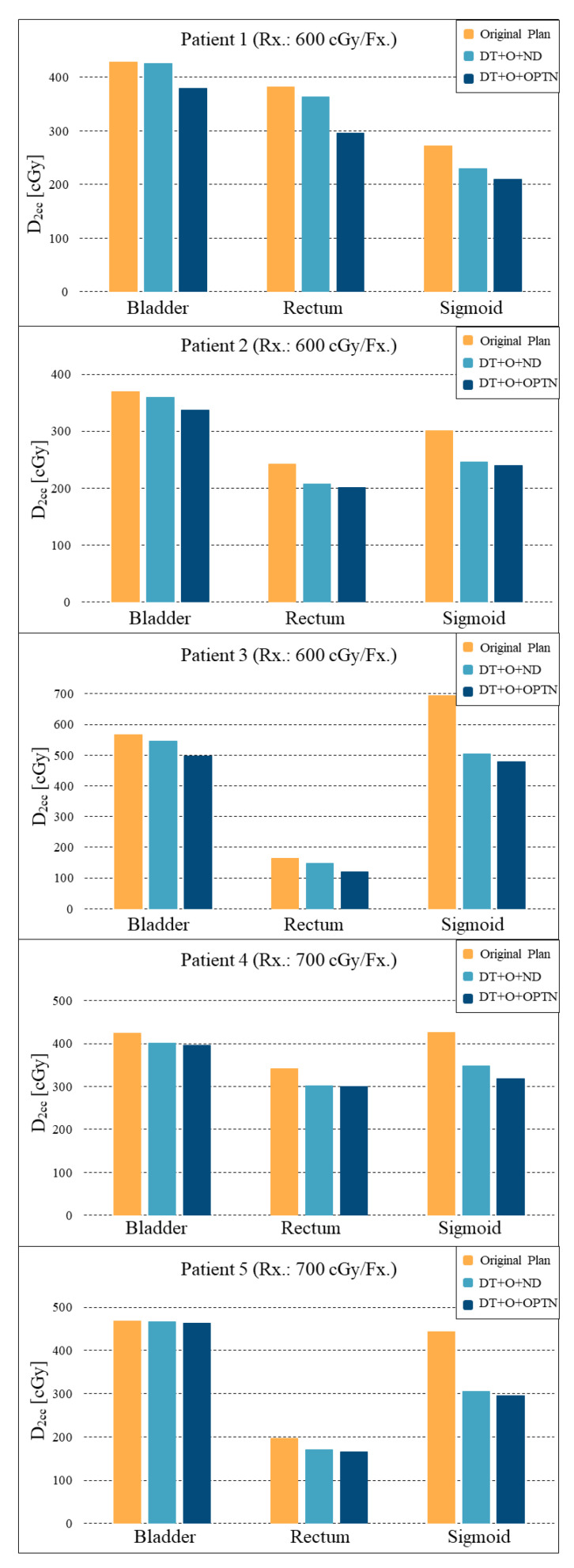
Comparison of D_2cc_ values achieved by each respective plans: original plan (yellow), DMBT + O + ND (light blue), and DMBT + O + OptN (dark blue). DMBT + O + ND: the original ovoids (O) and needles (ND) with the conventional tandem replaced by the DMBT tandem, DMBT + O + OptN: the original ovoids with the DMBT tandem and the needles spatially optimized.

**Table 1 jcm-11-06989-t001:** The prescription dose, CTV_HR_ volume, and D_90_ values between the original, DMBT + O + ND, and DMBT + O + OptN plans.

		Volume of CTV_HR_ [cc]	Rx. Dose [cGy]	D_90_ of CTV_HR_ [cGy]
Patient 1	Original Plan	58.8	600.0	567.4
	DMBT + O + ND			568.4
	DMBT + O + OptN			572.6
Patient 2	Original Plan	51.2	600.0	608.2
	DMBT + O + ND			613.1
	DMBT + O + OptN			610.1
Patient 3	Original Plan	45.3	600.0	537.0
	DMBT + O + ND			537.9
	DMBT + O + OptN			538.3
Patient 4	Original Plan	53.5	700.0	684.2
	DMBT + O + ND			681.7
	DMBT + O + OptN			684.5
Patient 5	Original Plan	59.7	700.0	686.0
	DMBT + O + ND			684.6
	DMBT + O + OptN			689.5

DMBT: direction modulated brachytherapy, O: original ovoids, ND: needles, DMBT + O + OptN: the original ovoids with the DMBT tandem and the needles spatially optimized.

## Data Availability

Not applicable.

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
