# Peer review of "A Novel Workflow with a Customizable 3D Printed Vaginal Template and a Direction Modulated Brachytherapy (DMBT) Tandem Applicator for Adaptive Interstitial Brachytherapy of the Cervix"

_jcm, 2022, doi:10.3390/jcm11236989_

Round 1

Reviewer 1 Report

This work demonstrates a clinical method of brachytherapy using a direction-modulated brachytherapy tandem applicator (DMBT) in combination with a patient-specific, 3D-printed vaginal needle track model for advanced image-guided adaptive interstitial brachytherapy of the cervix of the uterus.

A comparative study, between the OPs,  DMBT+O+ND, and DMBT+O+OptNs, of D2cc, was made for organs at risk (bladder, rectum, and sigmoid) to evaluate the quality of this method,

The work is very interesting for the aspect of clinical practice,  but I would like to know :

1-  So, in your dose calculation method, did you use the Acuros algorithm or MBDCA? please show it clearly in the text

2- You reported that there is a lack of a needle position optimization algorithm in your TPS, which requires you to manually optimize needle positions. It is necessary to explain this optimization.

Author Response

Dear Reviewers,

We appreciate your time and efforts to make this article better for readers. All points have been addressed and modified into the new manuscript. Thank you.

1-  So, in your dose calculation method, did you use the Acuros algorithm or MBDCA? please show it clearly in the text

-> We have used the Acuros dose calculation algorithm. This has been added in L75-65, L84-94 and L167-168.

2- You reported that there is a lack of a needle position optimization algorithm in your TPS, which requires you to manually optimize needle positions. It is necessary to explain this optimization.

-> A whole new paragraph describing the manual optimization has been added in L280-291.

Reviewer 2 Report

I would like to thank the authors for drafting this manuscript. I must admit that it was written in an unconventional fashion from a clinician's perspective. I have the following suggestions:

* Figure 1 legend is repeated in lines 82-92 and should be removed. 

* I would recommend reporting the Results section (line 232) and Discussion sections (starting at line 256) separately. i.e., start Results on line 232 and start Discussion on line 256.

* While a limitation paragraph is not present, the authors mentioned limitations of the current work throughout the manuscript. I would suggest adding a limitation stating the current workflow is based on a small number of cases. Another optional limitation to include would be the potential impact on the cost (time & money).

Author Response

Dear Reviewer,

We appreciate your time and efforts to make this article better for the readers. We have modified all your questions/comments as following:

1. Figure 1 legend is repeated in lines 82-92 and should be removed. 

-> Deleted.

2. I would recommend reporting the Results section (line 232) and Discussion sections (starting at line 256) separately. i.e., start Results on line 232 and start Discussion on line 256.

-> Split into two sections per your request.

3. While a limitation paragraph is not present, the authors mentioned limitations of the current work throughout the manuscript. I would suggest adding a limitation stating the current workflow is based on a small number of cases. Another optional limitation to include would be the potential impact on the cost (time & money).

-> We have re-organized a few paragraphs in the Discussion, especially at the last paragraph (e.g., "This workflow, while straightforward, may require some work and additional cost to start implementing.").
